# Effect of Thermo-Mechanical Treatment on the Microstructure Evaluation and Mechanical Properties of Fe-20Mn-12Cr-3Ni-3Si Damping Alloy

**DOI:** 10.3390/ma12071119

**Published:** 2019-04-04

**Authors:** Hyunbo Shim, Changyong Kang

**Affiliations:** 1Heavy Plate R&D Team, Hyundai Steel, Dangjin 31719, Korea; shim.hb0103@gmail.com; 2Department of Metallurgical Engineering, Pukyong National University, Busan 5991, Korea

**Keywords:** thermo-mechanical treatment, damping alloy, αʹ-Martensite, ε-Martensite, TEM, EBSD

## Abstract

This study was carried out to investigate the effect of thermo-mechanical treatment on the microstructure of Fe-20Mn-12Cr-3Ni-3Si damping alloy. Dislocation, αʹ, and ε-martensite were formed by thermo-mechanical treatment. The intersections of the surface relief and specific direction due to martensitic transformation were generated by thermo-mechanical treatment. They were then reversed to austenite with an ultra-fine grain size of less than 5 μm by annealing treatment at 700°C for 20min. The volume fractions of dislocation, αʹ, and ε-martensite were increased with the cycle number of thermo-mechanical treatment. In five-cycle number thermo-mechanical treated specimens, more than 45% of the volume fraction of ε-martensite and less than 3% of the volume fraction of α΄-martensite were attained. Therefore, in this article, the effect of thermo-mechanical treatment is briefly introduced, and these phenomena are explained in terms of the grain refinement of austenite, αʹ, and ε-martensite distribution and homogeneous dislocation distribution.

## 1. Introduction

Various types of human and material loss are caused by noise and vibration. Therefore, research interest has been increasing with respect to the requirements and regulations for damping noise and vibration. To date, the system, structure, and material damping methods have been used as the industrial noise and vibration prevention methods; however, the system and structural damping methods exhibit limited industrial applications because of the low damping capacity and applicability issues. Therefore, an effective material damping method that uses a damping alloy with high internal friction coefficient was investigated because of its excellent working property and adhesiveness, and related research was actively conducted [1,2,3,4,5,6,7,8,9,10,11,12,13,14,15].

The damping alloy converts the majority of the external vibration energy into heat or other types of energy due to its considerable internal friction [2]. Because the damping capacity of the damping alloy can be attributed to the movement of various defects, such as the potential that exists inside the alloy due to external stress, it is largely affected by the microstructure, which changes because of the thermo-mechanical history of the alloy, among other reasons [5,12]. Furthermore, the damping capacity of the alloy that causes stress induced martensite transformation is mainly dependent on the crystalline structure of the produced martensite [12,14]. Additionally, it is known that the damping materials function better when the strength, processability, frequency-dependency based on damping, and temperature dependency are low [2]. Further, the high strength material decreases the damping capacity [8]. Because it is difficult to develop a damping alloy that can satisfy these properties, only a few types of damping alloys are currently commercially available. Thus, even though various studies related to the development of the damping alloy that satisfy the required properties and offer high applicability as a structural material by combining both the strength and the damping capacity have been conducted [4,5,6,7,8,9,10,11,12,13,14,15,16,17,18], further research is required, particularly for attaining data to develop a damping alloy that offers an excellent combination of the strength and the damping capacity. Further, it is necessary to investigate the deformation behavior of the martensite produced by thermo-mechanical treatment, which is a useful method as a reinforcement method for austenite [9].

This study intended to acquire the data required for a damping alloy that exhibited both excellent damping capacity and high strength. Therefore, this study designed a damping alloy using the Fe–20Mn–12Cr alloy, which is renowned for its excellent damping capacity and strength [4]. In the study, the alloy’s damping capacity was increased by reducing the stacking faults and adding 3% Ni and 3% Si by considering the strength improvement resulting from solution hardening. The next section presents an analysis of the formation behavior of martensite (α′ and ε), which significantly affected the mechanical characteristics and the damping capacity after the annealing process, which was one of the methods used for improving the strength of the alloy.

## 2. Materials and Methods 

### 2.1. Specimens

To prepare the specimens, ingot was created using the chemical composition presented in Table 1 in a high-frequency vacuum melting furnace (Auto Tech, Busan, Korea). Hot rolling was performed at 1200 °C, and steel plates with 10 mm thicknesses were created. One way to achieve mixed structures consisting of austenite and stress induced martensite is the hot rolling process. Furthermore, these plates were acid-cleaned and cold rolled to create plates with various thicknesses and were considered to be the specimens for this test.

### 2.2. Thermo-Mechanical Treatment

The cold rolling ratio of 16% was set between the highest tensile strength with the room-temperature and damping capacity for performing the thermo-mechanical treatment [11]. Further, to reverse transform the majority of the martensite created by the cold rolling, the annealing treatment was applied to the martensite, which was maintained at 700 °C for 20 minutes, followed by water cooling. Therefore, the martensite was transformed into austenite by the annealing treatment [6,10]. Furthermore, up to five cycles of this thermo-mechanical treatment were performed, with one cycle for the cold rolling and the annealing treatments. Meanwhile, the specimens with various thickness values ranging from 1 to 2.5 mm were subjected to as many cycles of thermo-mechanical treatment as necessary to ensure that all their thicknesses became 1 mm. One cycle of the thermo-mechanical treatment process is shown in Figure 1 as a schematic diagram.

### 2.3. Observation of the Microstructure

The microstructure of the specimens before and after performing the thermo-mechanical treatment was observed using an optical microscope, SEM (Rigaku, Busan, Korea), and TEM (Hitachi, 200kV, Pohang, Korea). The observation was performed using TEM by processing the specimens to form 3 mm diameter thin plates after electrolytic polishing. Furthermore, to perform a detailed evaluation of the structures deformed by the thermo-mechanical treatment, the crystalline structure and finite structure information was identified by electron backscattered diffraction (EBSD, Rigaku, Pohang, Korea) with respect to the specimens’ image quality (IQ) maps and phase maps, among others.

### 2.4. X-Ray Diffraction Test (XRD)

The volume fraction of the microstructure after performing the thermo-mechanical treatment was measured using the cross-section area value of the peak that corresponded to each phase from the diffraction line acquired from the diffraction test at a speed of 2°/mm between temperatures of 20° and 100° using the Cu–Kα X-ray (Rigaku D/Max-Ⅱ A, Pohang, Korea) at room temperature [17]. For example, in the case of the ε-martensite volume fraction, the cross-section area under ε (100), ε (101), ε (102), and ε (103) peaks were calculated. The same method was used for the volume fraction of austenite.

## 3. Results

### 3.1. Microstructure

Figure 2 depicts the microstructure of the Fe–20Mn–12Cr–3Ni–3Si damping alloy using an optical microscope, denoting a small amount of martensite in the austenite structure with some twin crystals.

### 3.2. Change in the Microstructure by Thermo-Mechanical Treatment

Figure 3 denotes the microstructure of the specimen, which was treated with 16% cool rolling, using an optical microscope and a scanning electron microscope. Because of the deformation of some austenite into martensite after performing the treatment, which is denoted in Figure 3a, a larger amount of martensite existed than that shown in Figure 2. Further, Figure 3b denotes that martensite had a specific direction, and that some parts showed intersections or surface relief [10]. 

Figure 4 depicts the microstructure of the 16% cool-rolling-treated specimen using the scanning electron microscope to identify the martensite from the Fe–20Mn–12Cr–3Ni–3Si damping alloy after performing the treatment. Figure 4a denotes the dark field image of the microstructure, and Figure 4b shows the diffraction pattern (DP) from the phases in the dark field image and its analysis results. Based on these results, the ε-martensite in the hcp crystalline structure, the austenite in the fcc crystalline structure, and the α′-martensite in the bct crystalline structure were observed to co-exist. Thus, some of the austenite was deformed to α′-martensite and ε-martensite after the treatment [9]. 

Figure 5 depicts the microstructure of the thermo-mechanical treated specimen using the scanning electron microscope with the annealing treatment, in which the alloy was 16% cool-rolled, maintained at 700 °C for 20 minutes, and water cooled. Figure 5a denotes the bright field image of the microstructure, and Figure 5b denotes the selected area DP of the phase on the bright field image and its analysis results. Further, it could be determined that the potentials introduced by the thermo-mechanical treatment were ultra-finite austenite in a single-phase structure. Therefore, the majority of the martensite produced by the treatment was reverse-transformed by the annealing treatment to form finite austenite with a size of 5 μm or smaller.

Figure 6 denotes the microstructures of the specimens in the IQ maps of EBSD, which were one cycle and five cycles, to identify the structural changes after the thermo-mechanical treatment was performed. Several potentials were introduced owing to the thermo-mechanical treatment, and an increase in the number of cycles reduced the size of crystals. Further, the size of the finite austenite in a specimen that was subjected to thermo-mechanical treatment in five cycles was smaller than 5 μm. Therefore, as the number of thermo-mechanical cycles increased, the grain growth became difficult due to an increase in the inducing stress. This structural refinement was believed to be the main factor controlling the damping capacity.

### 3.3. Effect of Thermo-Mechanical Treatment on the Formation Behavior of Martensite

At room temperature, the treatment of steel with the austenite structure produced α′-martensite and ε-martensite, and an increase in the amount of treated specimen increased the amount of ε-martensite until ε-martensite was deformed into α′-martensite. Subsequently, the amount of ε-martensite decreased, whereas the amount of α′-martensite rapidly increased [11,12]. Further, the mechanical property of steel exhibiting such a deformation was largely dependent on α′-martensite, and its damping capacity was based on ε-martensite [12]. The formation behavior of martensite was crucial for determining the mechanical property and damping capacity of the alloy. Therefore, this study evaluated the formation behavior of martensite after performing thermo-mechanical treatment. 

Figure 7 denotes the microstructures of the specimens in TEM, which were subjected to three and five cycles of thermo-mechanical treatment to determine the effect of thermo-mechanical treatment on martensite formation. Further, both the specimens had ε-martensite and austenite structures with several potentials. 

Figure 8 depicts the analysis results of the diffraction lines acquired from the diffraction test of the specimen without performing the thermo-mechanical treatment (a), those of the diffraction lines acquired from the diffraction test of the specimen after performing one cycle of thermo-mechanical treatment (b), and those of the diffraction lines acquired from the diffraction test of the specimen after performing five cycles (c) to determine the change in the microstructure of the specimens after conducting the thermo-mechanical treatment using an X-ray diffraction test. While austenite and ε-martensite were identically detected in all three specimens, α′-martensite was not detected. Furthermore, as more cycles of thermo-mechanical treatment were performed, the higher the peak corresponding to ε-martensite was, which indicated an increase in the amount of ε-martensite.

Figure 9 denotes the microstructures of the specimen in the phase map after one cycle and three cycles to determine the formation of α′-martensite and ε-martensite by thermo-mechanical treatment from different perspectives. Figure 9a, which denotes the result after performing one cycle of thermo-mechanical treatment, showed the austenite structure and the ε-martensite structure (yellow), and α′-martensite was not observed. However, in a specimen that was subjected to thermo-mechanical treatment in three cycles, both austenite and ε-martensite, as well as a very small amount of α′-martensite (red), were observed. Furthermore, the size of the crystals was smaller in the case of the specimen subjected to three cycles of thermo-mechanical treatment than that for the specimen subjected to one cycle. Meanwhile, the TEM observation results in Figure 7 and the diffraction test results in Figure 8 could not be used to verify the existence of α′-martensite. However, the phase map results denoted a small amount of α′-martensite because the measurement sensitivity of EBSD with respect to α′-martensite was relatively superior to that of XRD or any other methods [10]. 

Figure 10 denotes the volume fraction based on the XRD test and the phase map of EBSD after the specimens were subjected to a different number of thermo-mechanical treatment cycles to determine the change in the volume fraction of each phase based on the thermo-mechanical treatment. As the number of cycles increased, the amount of austenite began to decrease until it increased slightly after five thermo-mechanical treatment cycles, whereas the amount of ε-martensite initially increased and slightly decreased after five cycles [11]. Additionally, α′-martensite was not detected in the original specimen or the specimen with the thermo-mechanical treatment in one cycle until the treatment cycle was performed, and after two cycles or more, an extremely small amount of α′-martensite began to be produced. Further, as the number of thermo-mechanical treatment cycles increased, it also gradually increased. However, the ratio of specimens subjected to five cycles of thermo-mechanical treatment was small and was equal to or lower than 3%.

Further, thermo-mechanical treatment produced α′-martensite and ε-martensite, and the increase in the number of thermo-mechanical treatment cycles increased their produced amount. Such results could be attributed to the increase in stress caused by the thermo-mechanical treatment that produced several potentials and stacking faults, which were good locations for generating the nucleus of martensite; they not only promoted the formation of α′-martensite and ε-martensite but also left some martensite. This prevented the reverse deformation of the martensite produced by the stress originating from the thermo-mechanical treatment to form austenite during the annealing treatment [10]. Therefore, the thermo-mechanical treatment increased the stacking fault energy, which was the transformation nucleation site of the martensite. As a result, it was found that fine martensite was formed, and these exhibited grain refinement effects, thereby improving the damping property [6,18].

## 4. Conclusions

The analysis of the microstructure of the Fe–20Mn–12Cr–3Ni–3Si damping alloy after conducting the thermo-mechanical treatment and the reverse-deformed annealing treatment achieved the following results:

(1) the thermo-mechanical treatment produced several potentials, α′-martensite, and ε-martensite, among others; 

(2) martensite produced by the thermo-mechanical treatment exhibited surface relief and specific direction; 

(3) the thermo-mechanical treatment produced ultrafine austenite whose size was smaller than 5 μm;

(4) α′-martensite was produced after the thermo-mechanical treatment cycle was increased by more than two; and 

(5) as the thermo-mechanical treatment cycle increased, the amount of potentials, α′-martensite, and ε-martensite increased. In addition, approximately 49% of ε-martensite was observed in the specimen subjected to five cycles of thermo-mechanical treatment when compared to less than 3% of α′-martensite.

## Figures and Tables

**Figure 1 materials-12-01119-f001:**
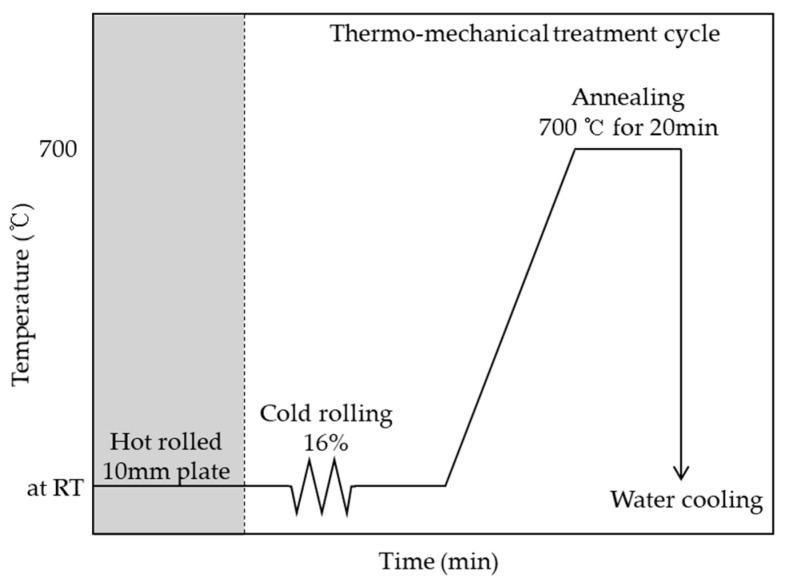
Schematic diagram of thermo-mechanical treatment cycle.

**Figure 2 materials-12-01119-f002:**
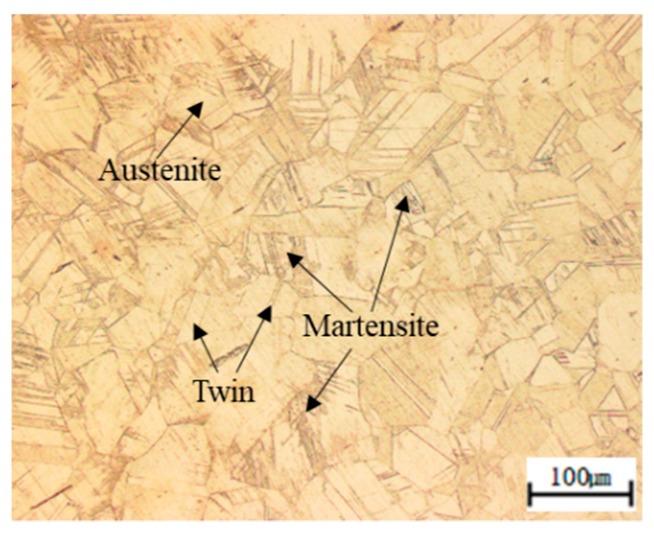
Optical micrograph of specimen hot rolled from 1200 °C; the base structure is austenite and a small amount of martensite is identified.

**Figure 3 materials-12-01119-f003:**
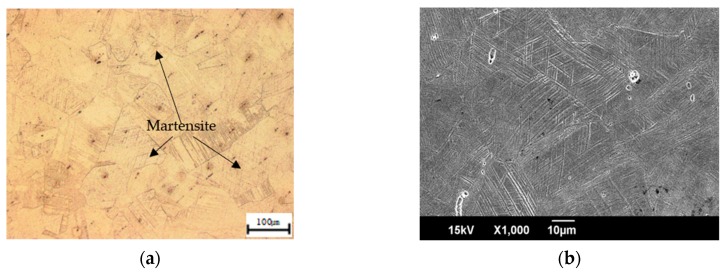
Micrographs of specimen cold rolled from 16%, listed as: (**a**) optical micrograph image; (**b**) SEM image. Note that micrographs are presented at different magnifications for clarity.

**Figure 4 materials-12-01119-f004:**
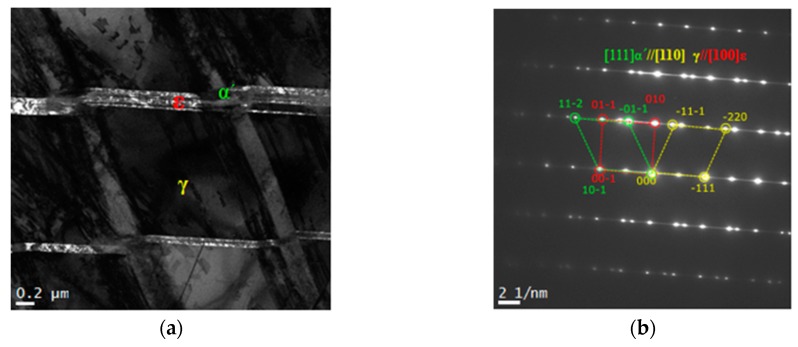
TEM micrograph of 16% cold rolled Fe-20Mn-12Cr-3Ni-3Si damping alloy, listed as: (**a**) dark field image; (**b**) self-aligned double patterning consisting of the [111] α′, [110] γ, and [100] ε zone axes taken from the images of (**a**).

**Figure 5 materials-12-01119-f005:**
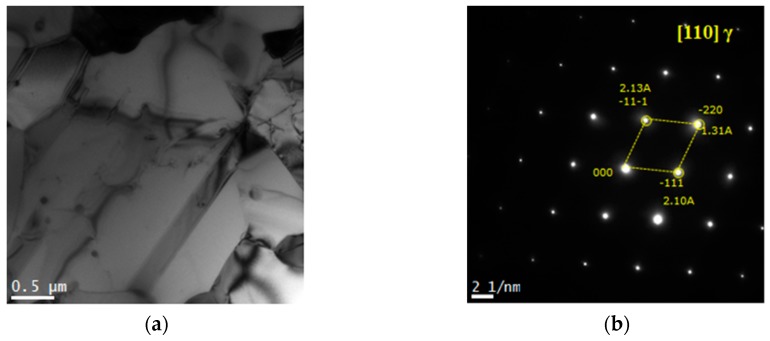
TEM micrograph of reversed austenite in 16% cold rolled Fe-20Mn-12Cr-3Ni-3Si damping alloy after annealing at 700 °C for 20 min, listed as: (**a**) bright field image of reversed; (**b**) self-aligned double patterning consisting of the [110] γ zone axes taken from the images of (**a**).

**Figure 6 materials-12-01119-f006:**
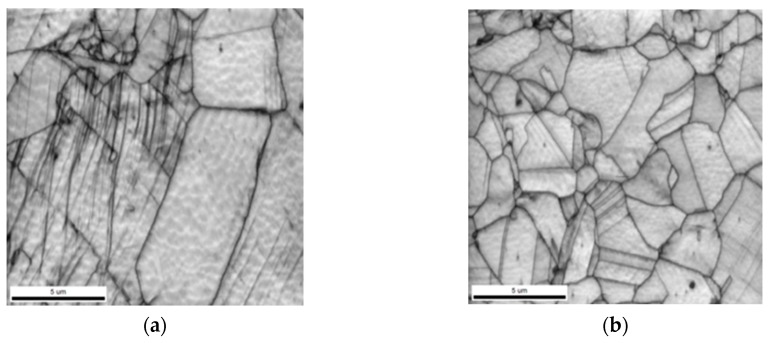
Image quality (IQ) map showing the effect of thermo-mechanical treatment in Fe-20Mn-12Cr-3Ni-3Si damping alloy, listed as: (**a**) one cycle; (**b**) five cycles.

**Figure 7 materials-12-01119-f007:**
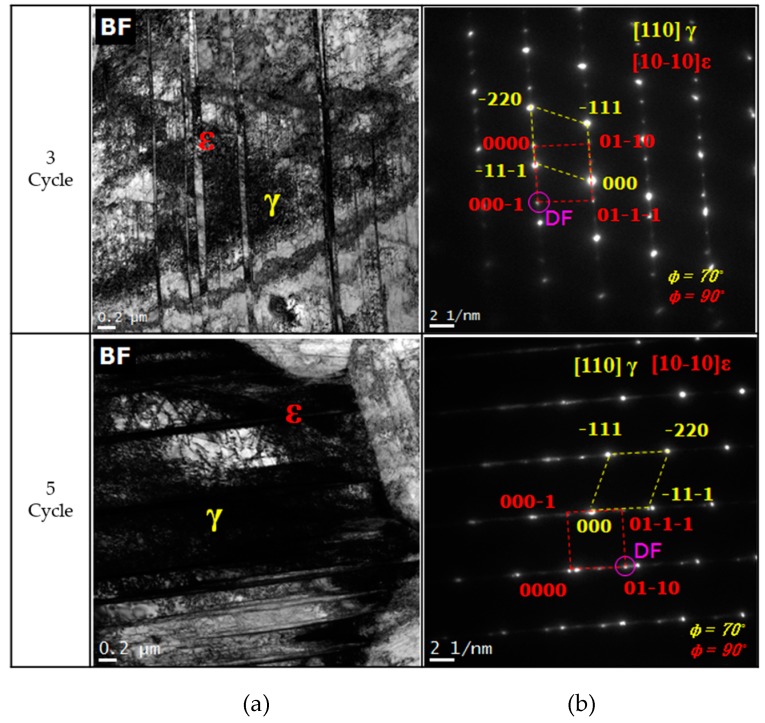
TEM micrographs showing the effect of the thermo-mechanical treatment in Fe-20Mn-12Cr-3Ni-3Si damping alloy, listed as: (**a**) bright field image; (**b**) self-aligned double patterning consisting of the [110] γ and [10-10] ε zone axes taken from the images of (**a**) each cycle of bright field image.

**Figure 8 materials-12-01119-f008:**
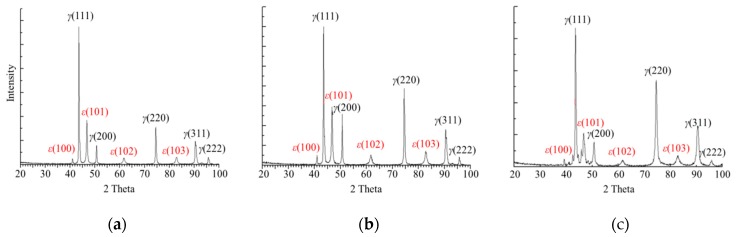
XRD patterns showing the effect of cycle number of thermo-mechanical treatment in Fe-20Mn-12Cr-3Ni-3Si damping alloy, listed as: (**a**) zero cycles; (**b**) one cycle; (**c**) five cycles.

**Figure 9 materials-12-01119-f009:**
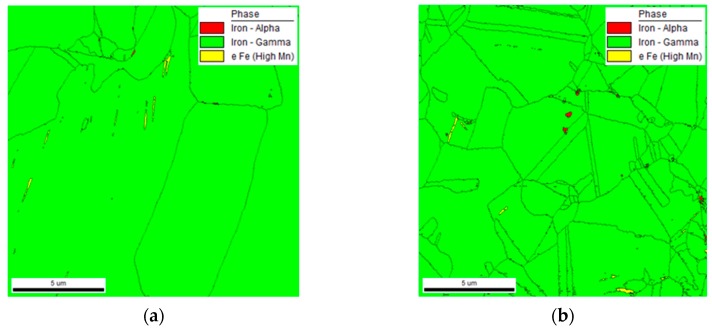
Phase map showing the effect of cycle number of thermo-mechanical treatment on the formation of α΄-martensite in Fe-20Mn-12Cr-3Ni-3Si damping alloy. High-resolution electron backscattered diffraction (EBSD) phase maps with image quality overlay, listed as: (**a**) one cycle; (**b**) three cycles.

**Figure 10 materials-12-01119-f010:**
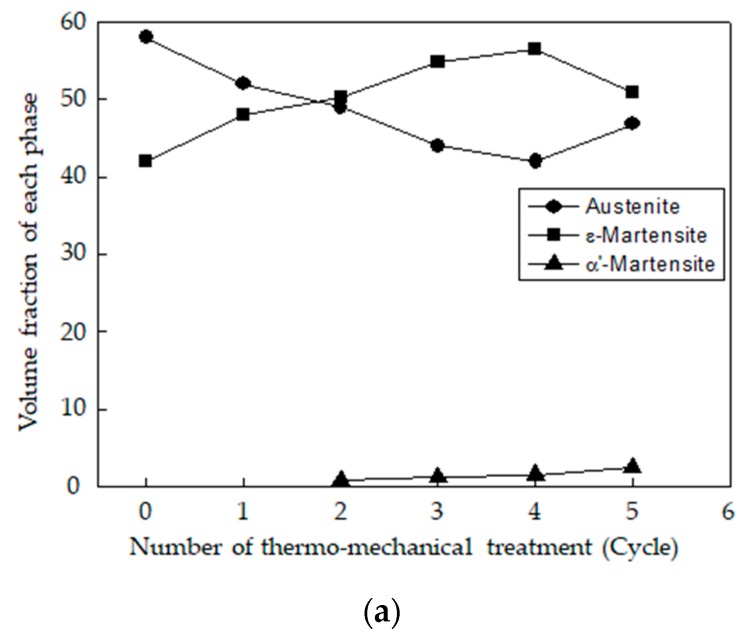
Effect of cycle number of thermo-mechanical treatment on the calculated volume fraction from XRD results of each phase in Fe-20Mn-12Cr-3Ni-3Si damping alloy.

**Table 1 materials-12-01119-t001:** Chemical composition of specimen (wt.%).

C	Si	P	S	Mn	Cr	Ni	Si	Fe
0.01	0.06	0.001	0.008	20,06	12.4	3.14	2.98	bal.

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
