# Peer review of "Effect of Thermo-Mechanical Treatment on the Microstructure Evaluation and Mechanical Properties of Fe-20Mn-12Cr-3Ni-3Si Damping Alloy"

_materials, 2019, doi:10.3390/ma12071119_

Reviewer 1 Report

The manuscript is presented in 8 pages and contains 15 literature references, 1 table and 8 figures, which is sufficient for qualifying works. The introduction and the methodology are well described, the references are not up-to-date. Please find new reference at last 3 years. In the manuscript there are two Figure 7. I think change the figure number.

This study designed a damping alloy using 51 the Fe–20Mn–12Cr alloy, which is renowned for its excellent damping capacity and strength, [4] and 52 it increased its damping capacity by reducing the stacking faults and added 3% Ni and 3% Si by 53 considering the strength improvement resulting from solution hardening. In the next section, this 54 study examines the formation behavior of martensite (αʹ and ε), which significantly affect the 55 mechanical characteristics and the damping capacity, after the annealing process, which is one of the 56 methods for improving the strength of the alloy.

Author Response

Response to Reviewer 1 Comments

Point 1: The manuscript is presented in 8 pages and contains 15 literature references, 1 table and 8 figures, which is sufficient for qualifying works. The introduction and the methodology are well described, the references are not up-to-date. Please find new reference at last 3 years. In the manuscript there are two Figure 7. I think change the figure number.

Response 1: Figure Number changed from 7 to 8.

Reviewer 2 Report

This is an interesting work on thermo-mechanical characterization of microstructure for Fe-20Mn-12Cr-3Ni-3Si Damping Alloy. The metallurgical features of the alloy have been validated by TEM and EBSD. However, to present a robust research article, the authors need to address the following comments.

1. Authors need to revise the abstract. Some words are repeated several times:

“Dislocation, αʹ and ε-martensite were formed by thermo-mechanical treatment. Martensite with surface relief and specific direction was formed by thermo-mechanical treatment. Martensite formed by thermo-mechanical treatment were…”

2. References [1-12] are cited at the end of paragraph 1. Please specify them in sentences, each reference is referring to which matter.  Similar mistake at Page 2, Line 45 (references [4-15]).

3. Page 1, Line 40: “high material strength” should be changed to “high strength material”

4. According to the title of the manuscript, the aim is to investigate the microstructure evolution of the material by the thermo-mechanical treatment. As a key role, the thermo-mechanical treatment inherently induces the grain refinement and precipitation through the microstructure. It seems that authors have neglected to explain this important characteristic of the thermo-mechanical treatment.

To show that the authors have a good understanding of the nature of the thermomechanical processing, this reviewer suggests this sentence with reference [*Ref] to be added to the second Paragraph (Page 1, Line 38):

“As a key role, the thermo-mechanical treatment inherently induces the grain refinement and precipitation through the microstructure, which can improve the microstructural and mechanical properties of the material [*Ref].” Additionally, it is known that…

[*Ref] Tamadon, A.; Pons, D.J.; Sued, K.; Clucas, D. Thermomechanical Grain Refinement in AA6082-T6 Thin Plates under Bobbin Friction Stir Welding. Metals 2018, 8, 375. https://doi.org/10.3390/met8060375  

5. It seems recent references haven't been used in the literature. This reviewer suggests the authors to update the references by adding some published after 2016.

Author Response

Response to Reviewer 2 Comments

Point 1: Authors need to revise the abstract. Some words are repeated several times:

Dislocation, αʹ and ε-martensite were formed by thermo-mechanical treatment. Martensite with surface relief and specific direction was formed by thermo-mechanical treatment. Martensite formed by thermo-mechanical treatment were…”

Response 1: Several times repeated words changed as follow.

Dislocation, αʹ and ε-martensite were formed by thermo-mechanical treatment. The intersections of these surface relief and specific direction due to martensitic transformation were generated by thermo-mechanical treatment. There were reversed to austenite with an ultra-fine grain size of less than 5 by annealing treatment at 700 for 20min.

Point 2: References [1-12] are cited at the end of paragraph 1. Please specify them in sentences, each reference is referring to which matter.  Similar mistake at Page 2, Line 45 (references [4-15]).

Response 2: Reference number notation was confirmed and revised.

 Ex) .[X] à [X]. , [X], à ,[X]  

Point 3: Page 1, Line 40: “high material strength” should be changed to “high strength material”

Response 3: The wrong word was corrected as follow.

There are changed from “high material strength” to “high strength material”

Point 4: According to the title of the manuscript, the aim is to investigate the microstructure evolution of the material by the thermo-mechanical treatment. As a key role, the thermo-mechanical treatment inherently induces the grain refinement and precipitation through the microstructure. It seems that authors have neglected to explain this important characteristic of the thermo-mechanical treatment.

Response 4: The title has been modified to include the evaluation content.

Effect of Thermo-mechanical Treatment on the Microstructure evaluation and mechanical properties of Fe-20Mn-12Cr-3Ni-3Si Damping Alloy

Point 5: It seems recent references haven't been used in the literature. This reviewer suggests the authors to update the references by adding some published after 2016.

Response 5: Added a reference as advice

13. Y. Watanabe, N. Iwata, H. Sato, Materials Science Forum 2017, 879, 101-106.

14. A. Tamadon, D.J. Pons, K. Sued, D. Clucas, Metals 2018, 8, 375.

15. S. Shin, M. Kwon, W. Cho, I.S. Suh, Mater. Sci. Eng. A 2017, 683, 187-194.
